
**The weather diary of Georg Christoph Eimmart for Nuremberg, 1695-1704**
Stefan Brönnimann[1,2]
[1]*Oeschger Centre for Climate Change Research, University of Bern, Switzerland*
[2]*Institute of Geography, University of Bern, Switzerland*
stefan.broennimann@giub.unibe.ch
**Abstract**
Weather diaries have long been used to reconstruct past climate. However, they could also be used to
reconstruct past weather. Weather reconstructions could help to better understand the mechanisms
behind, and impacts of, climatic changes. However, reconstructing the day-to-day weather requires
many diaries from different regions covering the same period, ideally combined with instrumental
measurements. In this paper, I describe the weather diary of Georg Christoph Eimmart from
Nuremberg, covering the period 1695 to 1704, which was particularly cold in Europe. The diary was
imaged from the Russian National Library in St. Petersburg and then digitized. It contains twice daily
weather conditions in symbolic form, wind direction, and information on precipitation and temperature
in text form. Symbols changed during the first two years, after which a much reduced (and stable) set
of symbols was used. Re-coding all days according to the later set of symbols, I find no signs of
inconsistency over time in symbols, wind direction, and precipitation information extracted from the
text. Comparisons with other sources confirm the day-to-day weather information in the diary. For
instance, the wind direction in Nuremberg agrees with the daily pressure gradient between Jena and
Paris. Three case studies further confirm the meteorological correctness of the information. This is
shown on behalf of an eight-day sequence of stormy weather in 1702, a study of the severe winter of
1697/8, and of the summer of 1695, which was cold and wet, possibly related to tropical volcanic
eruptions. The examples underline the consistency of the weather diary with other information and
suggest that weather reconstructions as far back as the late 17[th] century might become possible.
However, the spatial information is limited, and any approach arguably needs to make good use of the
temporal sequence of information.
**1. Introduction**
For decades, historians have used weather diaries to reconstruct past climate, i.e., to generate monthly
or seasonal index series (e.g., Pfister, 1999, see overview in Nash et al., 2021). However, they could
also be used to reconstruct past weather day by day. In fact, Manley (1975) described the daily
weather during the cold winter 1683/4 based instrumental data and weather diaries. Kington (1988)
and Lamb (1991) reconstructed daily weather types and drew daily weather charts for periods in the
18[th], 17[th], and even 16[th] century. They combined sparse observations with expert interpretation in a
reproducible way (Kington 1988).
As weather extremes and changes in weather have come into focus of climate science, reconstructing
daily weather is again considered an important goal. García-Herrera at al. (2007), Wheeler et al.
(2009) and numerous others have demonstrated the value of carefully reconstructing past extreme
weather events. However, to produce complete daily data series, automated and objective methods are
used rather than time consuming expert interpretations. Daily weather types (Schwander et al., 2017)
and gridded daily weather reconstructions (Imfeld et al., 2022) have been performed for Switzerland





back to 1763 and for Europe for the 1780s (Pappert et al., 2022). Other daily indices such as the wind
direction over the English Channel reach further back, to the late 17[th] century (Wheeler et al., 2010).
Cornes et al. (2012a,b) used daily data of sea-level pressure (SLP) from London and Paris to analyse
atmospheric circulation and storminess. There are many more examples for analyses of daily weather
300 years back (e.g., Brázdil et al., 2008, Filipiak et al., 2019), a review of approaches is given in
Brönnimann (2022). In addition to traditional statistical and numerical methods, new approaches such
as machine learning could possibly replace the expert approach pioneered by Manley, Kington, and
Lamb.
The success of any weather reconstruction approach ultimately depends on the available weather data.
From the turn of the 17[th] to the 18[th] century, several weather diaries are available. The diary from
Johann Heinrich Fries in Zurich covering 1684-1718 (Pfister, 1977) can be downloaded from EURO-
CLIMHIST (Pfister et al. 2017). The diary of the Kirch family in Leipzig (and Guben) and later
Berlin, covers 1677-1774 (Herbst, 2022) and was imaged by the author. Further diaries such as that of
David Grebner in Wroclaw covering 1692-1710 (Przybylak and Pospieszyńska, 2010) and Joseph
Dietrich in Einsiedeln covering 1670-1704 (Rohr and Schwarz-Zanetti, 2022) are under digitization
(not considered here). There are also a number of instrumental records from the late 17th and early
18th century (see Brönnimann et al., 2019a, Lundstad et al., 2022), many of which also have weather
descriptions. Combining all these data sets, reconstructing daily weather over Europe back to the late
17[th] century could become possible.
Here I add another weather diary, namely that of Georg Christoph Eimmart, founder of the first
Nuremberg (Nürnberg) astronomical observatory. His weather diary covers the years 1695-1704. This
paper describes the diary, its digitisation and (as the diary is mostly kept in symbolic form)
categorization. I then compare Eimmart's observations with other sources of daily weather
information.
The turn of the 17[th] to the 18[th] century is not only interesting as a test case as to how far back daily
weather reconstruction can reach, but it is also climatically interesting. It fell into the so-called "Late
Maunder Minimum" (Luterbacher et al., 2001), with particularly low temperatures in Europe
coinciding with low solar activity. At the same time, several volcanic eruptions (Hekla and Serua in
1693, Komagatake in 1694) might have affected climate (see Burgdorf, 2022).
The paper is organised as follows. Section 2 provides background about the observer and the diary.
Section 3 then describes the digitization and the data used for comparison. Results are presented in
Section 4. A brief discussion then follows in Section 5 and conclusions are drawn in Section 6.
**2. Georg Christoph Eimmart and his weather diary**
Georg Christoph Eimmart (1638-1705) is known as the founder of the first Nuremberg astronomical
observatory (the following text is based on Gaab, 2005, 2022). Eimmart attended the "Gymnasio
poetico" at Regensburg and enrolled in 1655 at the University of Jena, where he studied mathematics.
In 1658 he returned to Regensburg. After the death of his father, he moved to Nuremberg (following
his sister) around 1660. He worked as an engraver and got involved in the management of the
Academy of Painting founded in 1662, and from 1699 until shortly before his death he was director of
this institution.
Using the money earned with his artistic activity, Eimmart set up an observatory at Vestnertor, north
of Nuremberg Castle (Fig. 1) in 1678. At the end of the 17th century this observatory was the only
larger observatory in Germany. During special celestial events, the observatory was opened to the
public. In 1699, Eimmart was admitted to the Parisian Academy of Sciences, and at Leibniz's



suggestion two years later, he was also admitted as an external member of the Prussian Academy of
Sciences.
The written legacy of Georg Christoph Eimmart can be found today in the Russian National Library in
St. Petersburg. Vol. 40 of that legacy is the "Diarium tempestatum" (see also Gaab, 2022). It covers
the period 1 January 1695 to 25 November 1704 on 120 pages. The first three months are written in
ink, the remaining pages in pencil. The diary is structured in tables and contains only observations, no
measurements (although a sketch of an instrument is found on a verso page, Fig. 2; all other verso
pages are empty). The table is structured in columns, which on the first page are labelled as "Dies;
Qualitas Aeris; Temperam: cal: et frig:; Ventus". There is an apparent gap from 19-28 February 1700
("Incipit Dies 1. Martiy"), which is due to the change from the Julian to the Gregorian calendar.
Another (real) gap of four weeks occurs in January/February 1703.

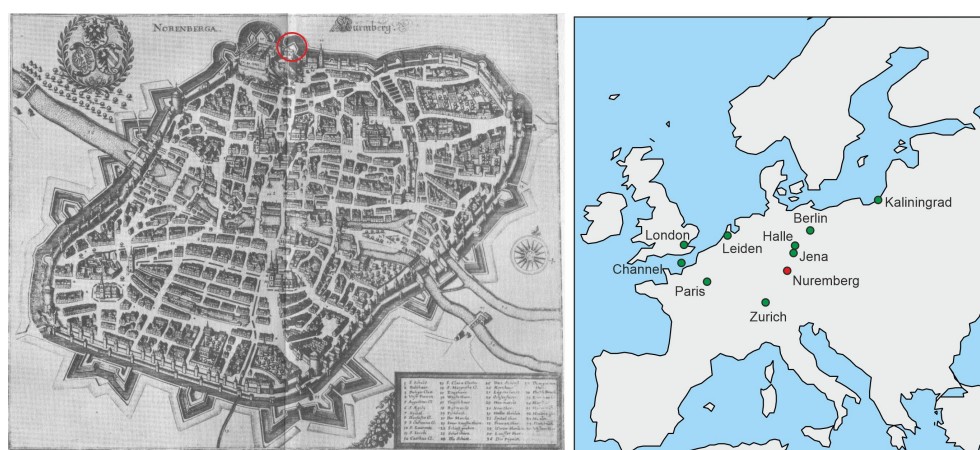


**Fig. 1.** (left) Map of Nuremberg, 1642 (engraving by Matthäus Merian from Germaniae, Edition Topographia
Franconiae, 1642; Wikimedia Commons, public domain) with the location of the observatory. (right) Location of
other weather observation series used in this study (dark green: instrumental measurements, light green:
observations).
An interesting aspect of the diary is that it is in its majority symbolic. This means that the entry is
mostly given as symbol, sometimes in words, sometimes both. In the first two years, each column (one
month, August 1695, has four columns, all others have three) has a distinct set of symbols that is used
(see Fig. 3). Particularly in the first two years, there are often several symbols in the same cell. After
2-3 years, however, the distinction between columns 1 and 2 gets lost, while the wind column remains
unchanged until the end of the diary. Some of the symbols, particularly those used in the second
column (on temperature), vanish and those used in the first column also appear in the second. A much-
reduced set of symbols is used in both columns after ca. 1697. There are mostly just three symbols:
full sun, upper half of sun, and long dash.
Wind is given in cardinal/intercardinal direction, plus a symbol probably referring to changing winds.
Sometimes several wind directions are mentioned in the same cell (see example in Fig. 2), which
might indicate several observations per day.
In addition to symbols and wind direction, there are also occasional latin expressions such as "nebula",
"pluvial per intervalla", or "nix". They describe phenomena related to precipitation (rain snow,
thunderstorm, lightning, rainbows), wind speed, and temperature. Occasionally other aspects are
mentioned (changeable weather, clarity of the sky). Words are sometimes written across both columns



(sometimes even also across the wind column), which leads to the hypothesis that, from ca. 1697
onward (after the same symbols are used in both columns), Eimmart wrote twice daily observations,
where the first column refers to morning and the second to afternoon. This is supported by the fact that
"tonitru" (thunders) appears only 6 times in the first, but 33 times in the second column. Likewise,
"aestus" (heat) appears 23 times in the first and 47 times in the second columns. However, there is no
text to prove this hypothesis.
The text entries are not independent of the symbols. In about half of the cases, text entries replace
symbolic entries, i.e., these fields then do not have a symbol on the sky conditions. For the other half
of cases with text entries these complement existing symbols. Sometimes the text refers specifically to
precipitation that has fallen during the night. At few instances, an additional row is even added
between two rows noting nocturnal rain or nocturnal storms, or this is made clear by subdividing cells.
Overall, there are slightly more text entries in the second column than in the first. Text entries in the
third column are rare.

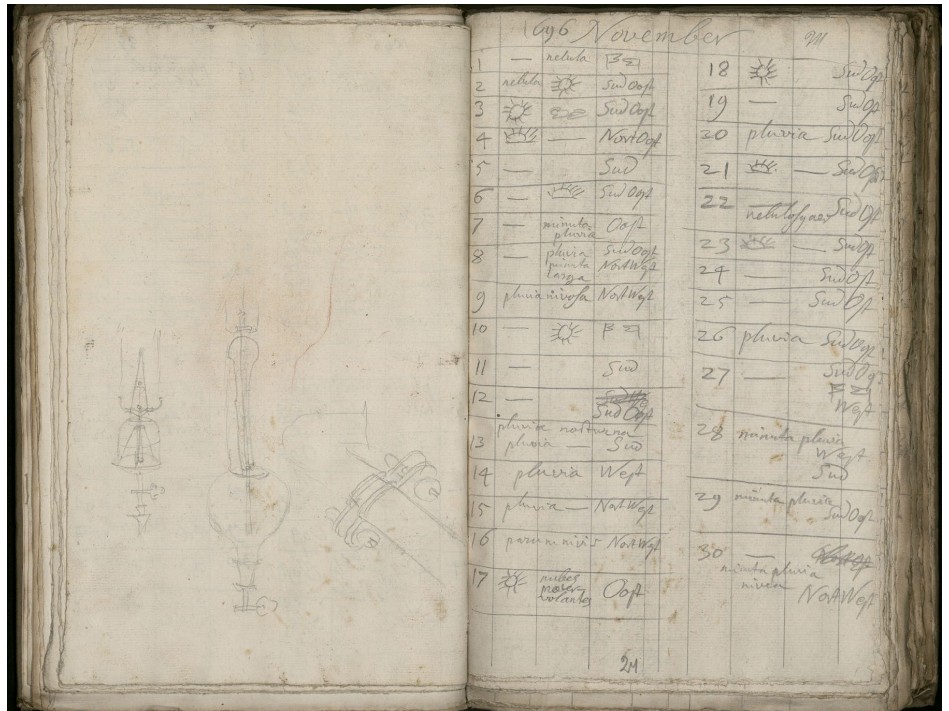


**Fig. 2.** Eimmart's diary for November 1696 from the National Library of Russia, St. Petersburg. The left page
shows sketches of scientific instruments (© National Library of Russia, St. Petersburg).

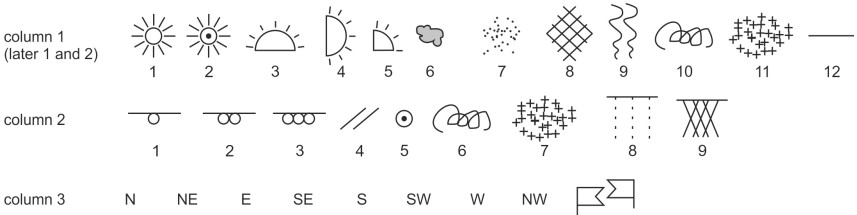


**Fig. 3.** Symbols used in the weather diary.





**3. Digitising and formatting the diary and comparison to other data sources**
Before digitizing, an inventory of all symbols appearing in the diary was compiled (Fig. 3), and a code
was assigned to each symbol. Then symbols were digitized as codes. For the text I use an additional
column. I transcribed the latin words as good as possible, some illegible words are marked with "$".
Words written over two fields are assigned to both fields with a corresponding bracket (e.g.,
„pluvius[„ and ]pluvius"; for the special case of August 1695, with 4 columns, I kept the three column
structure but assigned words across several columns only to one, with a note "[2 columns]"). A pipe
symbol "|" is used to indicate line breaks within a cell. In case of nocturnal weather, when the diary
makes it clear to which night this refers, I add it to the following day (the weather diary also mostly
does it this way) with a note "[previous night: ….]". Within the text, Eimmart sometimes uses a
symbol for the Sun, which I transcribed as "[solaris]".
From this raw transcription I generated several derived variables. First, I categorized the text entries
related to precipitation into snow ("nix", "nivigit" or similar), rain ("pluvia"), or rain and snow. In the
category rain I also included expressions such as "tonitru" (thunder) or "nebula pluviosa" (rainy fog),
but not "tempestas" (storm). To these I also added symbols 7 to 9 (rain) and 11 (snow) of Fig. 3,
column 1, as well as symbols 7 (snow), 8 and 9 (rain) of Fig. 3, column 2. Note that these almost
exclusively appear in the first 9 months. This variable is called precipitation.
Second, I formed a unified code based on the three main symbols used after 1697, namely "sunny"
(full sun), "partly sunny" (half sun), and "cloudy" (horizontal line). For this, I grouped symbols 1, 2
(column 1) and 5 (column 2) to the category "sunny", symbols 3 to 5 (column 1) as well as any
combination of a symbol 1-5 (column 1) with another symbol as category "partly sunny", and
categories 6 and 12 (column 1) to category "cloudy". All days for which the diary indicates a weather
symbol (76% in column 1, 60% in column 2) are thus assigned one of the three categories. This
variable is called "WeatherSymb".
Third, for most of the days with missing symbols, there is a text entry. In a further step, I also
generated a code for these entries. The terms "serenum" (clear) and "sunidy" (sunny) were coded as
"sunny", "coelum varium" (changeable), "pluvia per intervalla" (occasional rain), "pluvia minuta"
(little rain), "tonitru" (thunder), and symbol 8 (column 2) were coded as "partly sunny", and the terms
"pluvia", "pluvia tota dia", and "nebula" as well as symbol 9 (column 2) were coded as cloudy. In this
way, 97% (80%) of the days in columns 1 and 2, respectively, could be coded. The smaller amount in
column 2 is due to the fact that many of the text entries refer to temperature and winds. This variable
is called "WeatherSymbText".
Eimmart's diary entries were then compared with daily weather information from other sources (see
Fig. 1 for locations). This includes a weather diary from Zurich, pressure measurements from London
(Cornes, 2012a), Paris (Cornes, 2012b), Leiden, Halle, and Jena (Lundstad et al., 2022; see
Supplementary Material), temperature measurements from Paris (Rousseau, 2009, Pliemon et al.,
2022), Berlin, Halle, and Kaliningrad (Lundstad et al., 2022; see Supplementary Material), as well as
wind direction from ships on the Channel (Wheeler et al., 2010, Barriopedro et al., 2014). Note that,
first, many of the series only overlap partly with the Eimmart diary, thus limiting comparisons, and,
second, their quality is mostly unknown. For instance, the Halle records of both temperature and
pressure did not seem to be usable in their present form and were discarded. The following
comparisons address the internal temporal consistency, the temporal agreement of the diary with other
time series, the spatial consistency across Europe, and the consistency of daily wind direction with a
large-scale pressure gradient.



More comparability options would exist on a monthly scale by comparing with monthly
reconstructions. However, it is not straight forward to aggregate the daily weather information to a
monthly level, and the focus of this paper is on the daily scale. However, I compared the weather diary
with other documentary series (Burgdorf et al., 2022) and with monthly climate reconstructions for
two specific cases, namely the summer of 1695 and the winter of 1697/98. For these case studies I
used the ensemble mean of the reconstruction EKF400v2, which is a global, 3-dimensional climate
reconstruction based on data assimilation (Valler et al., 2022).
**4. Results**
*4.1. Time consistency of symbols, wind direction, and text*
In first step I analysed the frequency of each symbol per month, over both columns. Due to the large
number of symbols used, especially at the beginning, I ignored the temperature related symbols
(symbols 1-4, column 2), which only appear in the first 4 months, and grouped the symbols remaining
related to the general weather characteristics and the sky into four main categories (see Fig. 4). I then
counted each use of a symbol and divided it by the total number of symbols in that month. There
might be several symbols per day, or just one. Note also that text is ignored here.
The results (Fig. 4a) show what was already observed during the digitization, namely that in the
beginning, many different symbols appeared while later basically three symbols were used. The grey
category of symbols vanishes almost completely and from around 1698 onward, the use of the
symbols is rather consistent. Figure 4b shows the results obtained when simply ignoring they grey
category of symbols. Their frequency over time changes less, but many days have no category. The
corresponding figure for variable "WeatherSymb" (Fig. 4c) shows slightly different behaviour in the
first years compare to later. When also text entries are categorized (variable "WeatherSymbText" Fig.
4d), no obvious inhomogeneity is seen anymore. The figure resembles Fig. 4b, but now almost all days
have a category. Note that cloudy conditions are more frequent because most text entries concern rain.

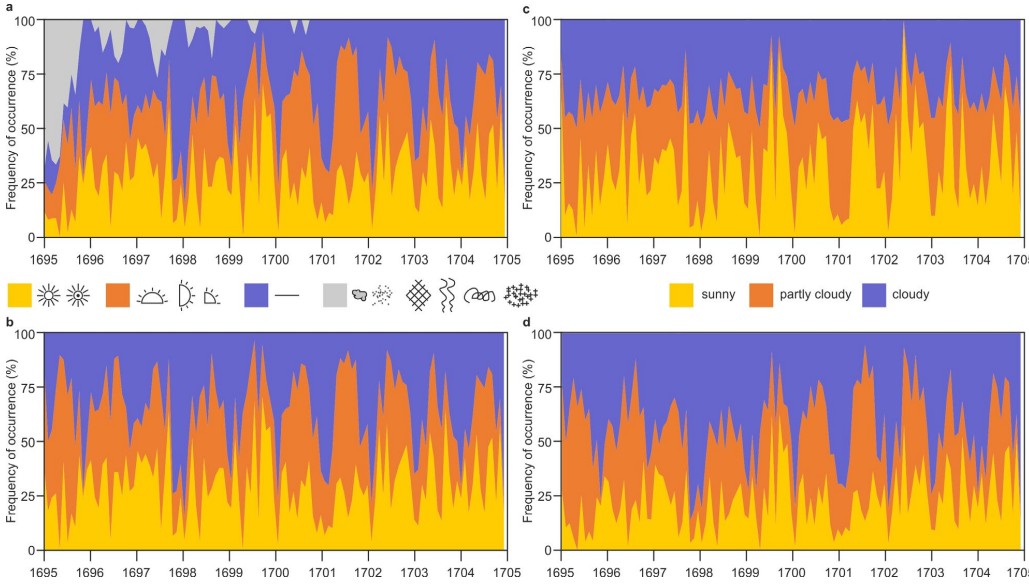

**Fig. 4.** Frequency of occurrence of weather symbols per month relative to all symbols in that month. **a** all
symbols (grouped into four categories), **b** only the first three categories, **c** variable "WeatherSymb", **d** variable
"WeatherSymbText".



Similar as for the weather symbols, I also analysed the wind direction. I counted the letters for the
cardinal directions (N, E, S, W as well as one letter for mixed), divided the counts by the number of
letters per day, and then averaged the number per month. Note that in this procedure NW counts as
two letters (each weighted half), N as one. This allows a first, albeit simple visualization (Fig. 5a).
Results show variations in occurrence, but no obvious inhomogeneity is seen in this plot.
The same was also done for the variable precipitation with its categories "snow", "rain and snow", and
"rain" (Fig. 5b). Variability in precipitation is high, but once again there is no evidence for an
inhomogeneity. The figure indicates a decrease of rain days over the 10 years. However, this might be
a true climatic signal. The large spike in the beginning of the series, in summer 1695, will be analysed
later and is arguably real.

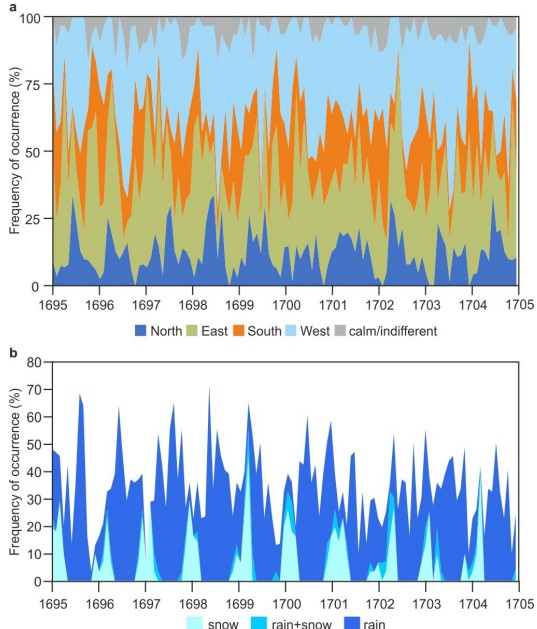


**Fig. 5.** Frequency of occurrence of **a** cardinal wind directions per month and **b** days with snow, rain and snow, or
rain.
*4.2. Analysis of the wind direction*
As a next step I compared the daily wind direction at Nuremberg with the daily SLP gradient between
Jena and Paris in the year 1702 (the only year for which data from Jena are available). A positive
gradient is expected to correlate with winds from the South or East, whereas a negative gradient is
expected to correlate with winds from the North or West. I therefore grouped the winds accordingly
(S, SE, E, SW vs. W, NW, N, NE). Most of the days had only one wind direction. If more than one
was noted, I excluded days that would fall into both categories.
To calculate the pressure gradient between Jena and Paris, I reduced the Jena data to sea level. As no
temperature information was available, I assumed a sinusoidal seasonal cycle of temperature varying
between -4 °C (in January) and 20 °C (in July). As the thus obtained SLP data were clearly too low in
Jena, I added the mean difference between Jena and Paris such that both series have the same mean. I
analysed the data using a contingency table and by stratifying the SLP difference according to the
wind direction.





The contingency table (Table 1) clearly shows a very strong association between pressure gradient and
wind direction in the expected sense. A statistical test (Fisher's exact t) confirms the high significance
of the results. However, while there are more cases with a negative than with a positive SLP difference
(note that the average difference is zero), there are clearly more cases with an easterly-to-south-
westerly wind than a westerly-to-north-easterly wind. The histogram of SLP difference (Fig. 6) also
clearly shows a difference in the SLP gradient depending on the wind direction. Note that deviations
are expected. Apart from the measurement errors and errors in the reduction to SLP, it should be noted
that the SLP gradient may not be the best proxy for wind. Furthermore, Paris is relatively far away,
and the wind over land is not geostrophic. Thermotopographic winds or topographically channelled
winds may overlay the large-scale flow. Wind is a variable with a high variability, and observations
provide only a snapshot. Note also that the time of day of observations is not known. In light of these
uncertainties, the clear presence of a signal is therefore encouraging.
**Table 1.** Contingency table of wind observations in Nuremberg and SLP differences between Jena and Paris. The p-value
refers to Fisher's exact t.

|  | E/SE/S/SW | W/NW/N/NE | Sum |
|---|---|---|---|
| ΔSLP>0 | 118 | 40 | 158 |
| ΔSLP<0 | 72 | 98 | 170 |
| Sum | 190 | 138 | $p<0.0001$ |


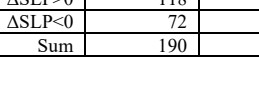

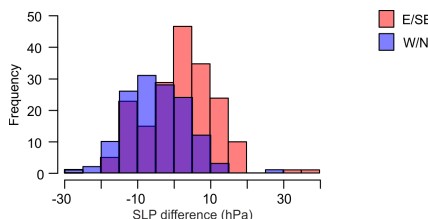


**Fig. 6.** Histograms of SLP difference between Jena and Paris stratified by the wind direction in Eimmart's diary.
*4.3. Case studies*
4.3.1. Weather maps for October/November 1702
Next I analysed the entries in Eimmart's diary together with all other available information for an 8-
day sequence in 1702. The sequence was chosen as a strong storm was noted in this period in the
observations from Kaliningrad in the night from 31 October to 1 November (Anonymous, 1703). Note
that the year 1702 is arguably the year with the best data coverage within the 1695-1704 period.
Instrumental data are available from Jena, Kaliningrad, Paris, London, Berlin and Kaliningrad,
weather observations from Jena, Kaliningrad, Zürich, Nuremberg, and the English Channel. For
display purposes, I deseasonalised (by fitting the first two harmonics of the seasonal cycle) and
standardized the instrumental data with respect to the year 1702 and expressed the anomalies in
standard deviations. For Berlin where the observation hours change rapidly, I only used the 8 days
displayed, chose the observation closest to 8 in the morning and standardized the temperatures.
The sequence of maps (Fig. 7) starts with mild temperatures, sunny or changeable weather and
moderately low pressure across Europe. The next day pressure increased in Paris and Jena, it remained
rather warm. Pressure remained low over Kaliningrad. The next three days saw a clear pressure
increase over London, Paris, and Jena, with westerly flow. Temperatures were high especially on 30
October. Pressure and temperature then both decreased on the day of the storm. Pressure increased
right after the passing of the storm, accompanied by a marked temperature decrease and mostly sunny
weather.



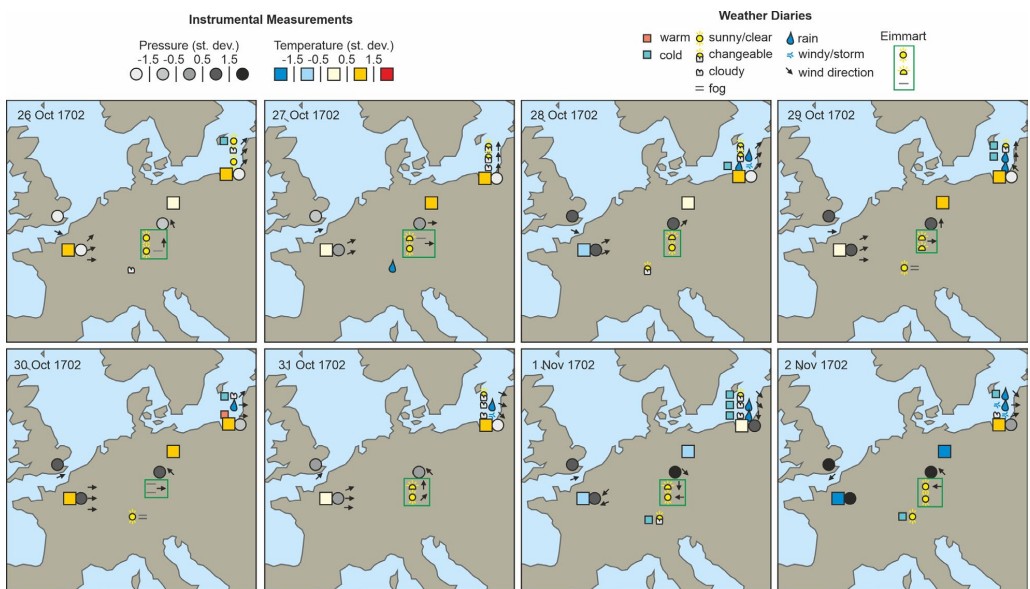


**Fig. 7.** Maps of pressure, wind, and temperature for an eight-day sequence in 1702. A vertical sequence of
symbols at the same location indicates sub-daily data (progressing from top to bottom).

While this sequence is plausible, the spatial information is insufficient to draw weather maps. There
could have been a quick succession of passing depressions over Northern Europe, that may have gone
unnoticed. However, the maps depict the high-pressure system to the south. Also, the passage of a cold
front on 1 Nov is clearly seen.
4.3.2. The harsh winter of 1687/8.
The second case study relates to the harsh winter of 1697/8. This winter is particularly well known in
England (Kington, 1999), where it was the coldest among six very cold winters in the 1690s.
However, the winter was also cold over Central Europe (Pfister and Wanner, 2021). A number of
European weather series exist on a daily scale for this winter, including pressure in London, Paris, and
Leiden, temperature in Paris, wind over the English Channel, cloud motion and cloud cover from
Paris, and weather observations from Zurich and Nuremberg. All observations are shown as time
series in Fig. 8.
Temperature measurements in Paris as well as notes on temperature in the Zurich series point to a
sequence of many cold spells during the entire winter and spring. For Nuremberg, notes on
temperature are sparser, but they also agree with the two other series. According to Kington (1999),
the first snow in London fell on 24 November. In Nuremberg and Zurich snowfall is reported on 22
November. We also see periods of higher temperature and thawing weather (or rainfall), such as in
early December. A snowy and cold period follows at all sites in mid-December. January then was
particularly cold in Paris, London, and Zurich. A notable pressure drop occurred in Paris on 1 February
1698, arguably associated with a warm front (Kington, 1999). Temperatures increased everywhere,
and precipitation fell as rain in Nuremberg (note that the temperature increase here occurred ca. two
days later than in Paris). The first half of March was again cold at all sites. Cold spells again occurred
in late April and early May. In fact, May 1698 still is the coldest May on record in the Central England
temperature series. This brief analysis shows that the daily series are consistent with each other. Even
wind directions agree well for locations close to each other (e.g., English Channel and Paris).

**Fig. 8.** Daily weather in the winter 1697/8. For Nuremberg, 1 to 3 marks the columns, subdivided into text
entries that were transformed into symbols (upper line) and symbolic entries (lower line).





As a further opportunity for comparison, I considered the chronicle by Johann Laurentz Bünti (Bünti,
1973) from central Switzerland, which points to snow fall on 3 and 8 May and again on 21 May
(«Hierauff folget den 3. May ein Schnee [...] Den 8.ten May hat man wiederum im ganzen Boden
Schnee.»). Bünti writes that precipitation was very high and that it fell as snow in the mountains, such
that there was more snow in the mountains in May 1698 than in many winters.
To further investigate this winter, I analysed temperature and pressure anomalies in the reconstruction
EKF400v2 (Fig. 9). The fields are expressed as anomalies from the subsequent 30-yr period. The same
figure also shows hand-drawn pressure maps by Wanner et al. (1995). These fields have no scale (and
are here compared with anomalies). However, analysing the position of highs and lows (or positive
and negative anomalies), I find a mostly good agreement, indicating that the data assimilation
approach and an expert approach give consistent results. Into these we can now embed the weather
diaries. The fields show that January and February were actually even colder in Central Europe than in
Western Europe. In January, Eimmart notes mostly easterly winds and a horizontal line, arguably
denoting a persistent stratus. These features stand out over the 10 year period and are also clearly seen
in Fig. 4 (increased blue area) and Fig. 5 (increased green area denoting easterly winds). This is very
well in line with the monthly charts shown in Fig. 9. The cold spells in May seemed to have had more
pronounced effects in Zurich than in nearby Nuremberg, consistent in the reconstructed fields, the
hand-drawn fields, and the diary entries.

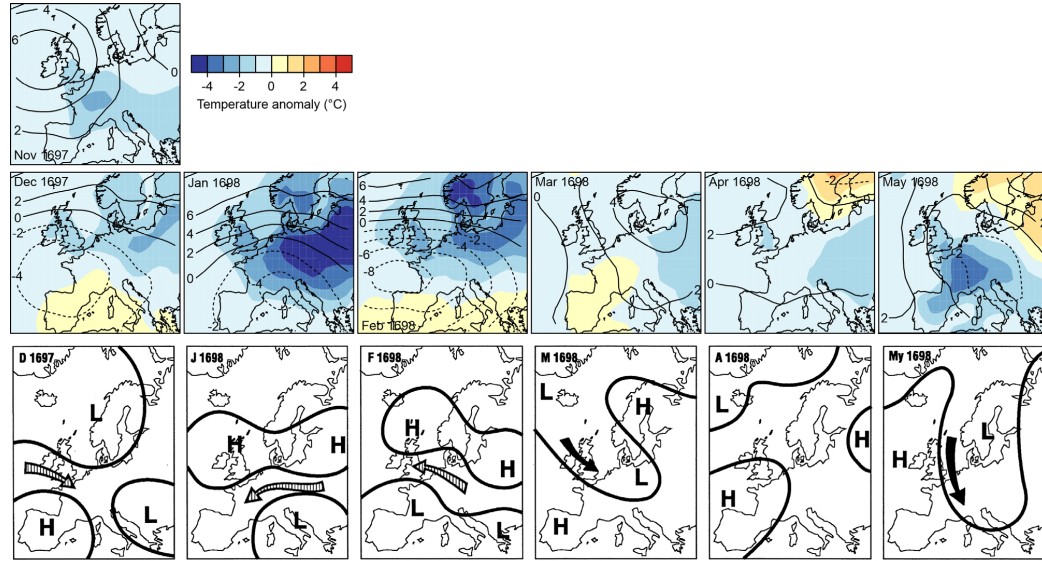

**Fig. 9.** Monthly temperature (colours) and pressure anomalies (contour, in hPa) relative to the period 1698/9 to
1727/28 from EKF400v2 (top) and an expert reconstruction of the pressure distribution and main flow for the
same months form Wanner et al. (1995).
4.3.3. The summer of 1695
Finally, I also analysed the summer of 1695, which is known as a very cold summer (July 1695 was
the second coldest July in the Central England temperature series behind 1816). The summer was also
cold in Switzerland. Bünti (1973) writes that 1695 was a late and wet year, with frequent summer
snowfall events in the Alps («Sonsten ist dissess 1695. Jahr ein spätes und nasses Jahr gesein; im
Summer [wurden] offt die Alpen überschnyt»).



For this case, I analysed the variables "Precipitation" and "WeatherSymbText" from Eimmart's diary
as well as temperature and weather conditions and in Zurich in June to August. I compared the
frequencies in 1695 to those obtained in 1696-1704 (Fig. 10). I also compared the result to
documentary climate data for that summer and to EKF400v2. For comparison, I standardized both
based on the 1696-1704 period (EKF400v2 is also shown as anomalies without standardizing).
Although this is a small sample, it is the longest statistical analysis the Eimmart diary allows.
The summer of 1695 was clearly less sunny in Eimmart's diary, while cloudy conditions were more
frequent. Also, there were more days with precipitation in Jun-Aug 1695 than in the average of all
other summers. A very similar behaviour is found in Zurich, where sunny conditions were less
frequent and "changeable" more frequent. A clear signal is also found in the temperature notes of
Zurich. Cold days were about twice as frequent in 1695 as in the reference period, mostly at the
expense of very warm days. For Nuremberg, there are too few temperature notes for an analysis.

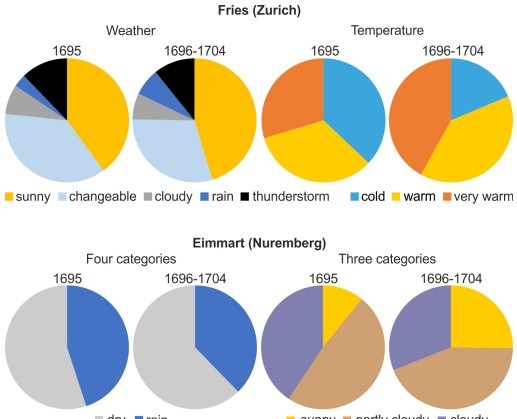


**Fig. 10.** Frequency of weather descriptions in the diaries of Eimmart (Nuremberg) and Fries (Zurich) for the
summer (Jun-Aug) of 1995 as well as for the remaining 9 summers of Eimmart's observation period.
The two diaries can now be compared to two further sources on that summer: documentary data (Fig.
11, top, Supplementary Material) and reconstructions (Fig. 11, bottom). Both sources show that it was
cold particularly in the northeast of the domain, less so in Western France and Spain. The
documentary data indicate no specifically cold period in Italy, whereas the reconstruction suggest
somewhat lower temperatures (note that the documentary data and reconstructions are largely
independent, only the monthly index series indicated in Fig. 11, top, with "JJA" were assimilated into
EKF400v2). Zurich and Nuremberg were both in a region that was affected by the adverse weather,
although not in the core region of this climatic anomaly.
**5. Discussion**
The weather diary by Georg Christoph Eimmart from Nuremberg, covering the period 1695 to 1704,
might be a useful addition to the compilations of existing weather diaries. The diary stands out in that
it is mostly symbolic for sky conditions, complemented with wind direction and text for precipitation
and temperature. Although there are quite large changes in the use of symbols and of text in the first 1-
2 years, derived variables that group sky conditions and precipitation each into 3 categories according
to both symbols and text show now sign of inhomogeneity. The same result is found for wind
direction. Observations seem to have been performed in a consistent manner over 10 years.



The diary compares well with other sources of information, such as the diary from Fries in Zurich. It
also compares well with instrumental data, as demonstrated by comparing wind direction with a large-
scale SLP gradient. This confirms that the diary not only has useful information on climate, but also
on the daily weather. Combining this diary with other available observation series shows that there is
spatial information in the weather data. Some features, such as the passing of a cold front, can be
clearly seen, but it is still difficult to draw detailed weather maps directly from this information.
Several more series will be available in the near future and might further help towards that aim.
However, there is also information in the time sequence of weather at each of these sites. Given the
sparse information in space, it is essential to also exploit the information in the time sequence. Novel
approaches such as deep learning algorithms might potentially be used for weather reconstruction but
arguably would have to make use of time sequences in order to be successful. Such methods first need
to be tested extensively in long data sets generated from recent products using tools such as synthetic
weather diaries (Brönnimann, 2021).

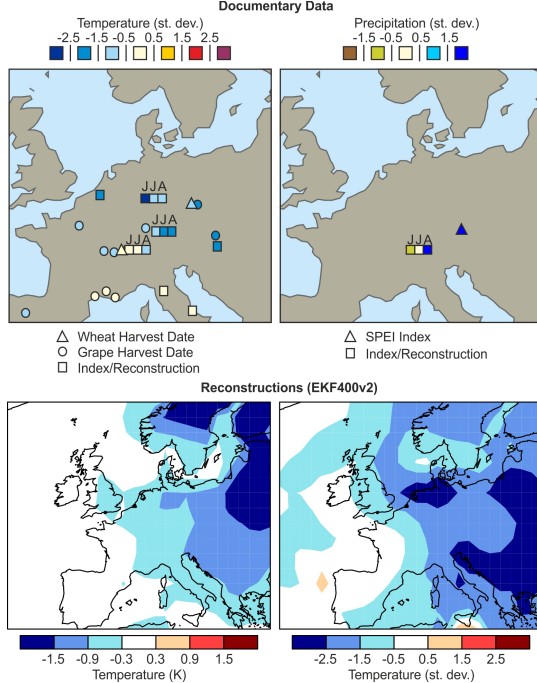


**Fig. 11.** Top: Standardised anomalies (relative to 1696-1704) of documentary data from Burgdorf et al. (2022)
for temperature and precipitation. Bottom: Anomaly (left) and standardized anomaly (right, both relative to
1696-1704) of temperature in EKF400v2. Note that EKF400v2 includes the four monthly series shown in the top
row (three temperature indices and one precipitation index), but not the other series.

Finally, the diary, together with other sources of information, provides some insights into the climate
processes in the 1690s, a period that was characterized by cold winters in Europe and also cold and
rainy summers. The winters such as 1697/98 were likely characterized by frequent blocking and a
meridionalisation of circulation. This is also seen in Eimmart's diary. The pressure difference between
Paris and London (Cornes et al., 2012a,b) exhibits its lowest values (winter average) in the 1690s (the
winters 1695, 1694, 1692, and 1698 occupy ranks 1, 2, 4, and 11). The North Atlantic Oscillation
index calculated from EKF400v2 also shows low values in these years (see Brönnimann, 2022).
These winters fell into the so-called Late Maunder Minimum (Luterbacher et al., 2001), when sunspot
activity was very low. In fact, these cold winters have often been attributed to low solar activity. A
more meridional circulation with more frequent blocking events due to low solar activity would be in
line with statistical analyses of later data (Woollings et al., 2010). In addition to solar forcing, also
volcanic eruptions cold have played a role. Eruptions occurred in 1693 (Hekla, Serua) and 1694
(Komagatake), and although winters following volcanic eruptions sometimes show a winter warming
in north-eastern Europe, this dynamical effect does not always appear and cold seasons following
volcanic eruptions may also be cold. The summer case (1695) shows the cooling expected following a
volcanic eruption (Raible et al., 2015); documentary data confirm this also on a hemispheric scale
(Burgdorf, 2022). Whether cold winters and cold summers in the 1690s are related remains to be
studied. There are several possible memory effects that might help to maintain the cooling from the
summer to the next winter and spring, including the oceans (Raible et al. 2015) and Eurasian snow
cover (Reichen et al., 2022). All factors together may have generated a decade of cold weather similar
to the early 19$^{th}$ century (Brönnimann et al., 2019b), when both summers and winters were cold
particularly over Eurasia. Historical weather diaries could help to further shed light on climate
mechanisms operating on a decadal scale related to volcanic eruptions and a solar minimum. However,
more weather and climate data are needed for this.
The first instrumental series in Nuremberg covers 1718-1730 (observer: Rost, 3-4x daily), another
series (observer: Doppelmayr, daily) covers 1732-1743 (see Brönnimann et al., 2019a). Both were
digitised and are included in the electronic supplement. Both also contain wind direction, such that a
1695-1743 record of sub-daily wind could be generated. However, homogeneity needs to be assessed.
**6. Conclusions**
This paper describes the digitization of the weather diary of Georg Christoph Eimmart. The diary
contains information on sky conditions (in symbolic form), precipitation, temperature, and wind in
Nuremberg, 1695-1704. It is relevant as the 1690s were a particularly cold decade in Europe. At the
same time, this approximately marks the period back to which daily reconstructions of weather might
be possible. The newly digitized diary might contribute towards this aim.
The diary structure changes during the first ca. two years, but afterwards (for wind throughout the
period), the diary is consistent. Comparisons with other series from Europe show that the diary
provides useful and usable information on the daily weather. For instance, the local daily wind
direction in Nuremberg agrees with the large-scale pressure gradient. The usefulness is further
demonstrated on behalf of several case studies, covering a storm passage in October/November 1702,
the cold winter of 1697/8, and the cold summer of 1695. These cases also show that the spatial
information at the daily level is inevitably sparse in the late 17$^{th}$ century. Any approach to reconstruct
the daily weather during this time arguably needs to make use of the temporal as well as the spatial
information.
**Funding Information:** The work was funded by the Swiss National Science Foundation project WeaR (188701)
and the European Commission through H2020 (ERC Grant PALAEO-RA 787574).
**Acknowledgements.** I would like to thank Andrey Martynov for his help in obtaining the diary, Juhyeong Han
for her help in digitizing it, and Yuri Brugnara for coordinating the digitising work. Hans Gaab and Klaus-Dieter
Herbst pointed me to the diary. I also thank Heinz Wanner for the hand-drawn reconstructions and Rolando
Garcia-Herrera for providing the wind data. The simulations underlying EKF400v2 were performed at the Swiss
Supercomputer Centre (CSCS).



**Supplement:** Supplementary files include the diary in xls, the data files for Jena, Kaliningrad, Leiden, and
Nuremberg (instrumental data) in the SEF format and xls, an ASCII file with the documentary data (Fig. 10), R-
Code and a readme file.
**Code Availability:** R-code to produce Figures 9 and 11 is given in the Supplement.
**Data availability statement:** The diary is in the Supplementary Material, EKF400v2 is available from
DKRZ/WDCC (doi:10.26050/WDCC/EKF400_v2.0), the Fries diary is available from EURO-CLIMHIST
(www.euroclimhist.unibe.ch), pressure data from Paris and London are available from the Climatic Research
Unit (https://crudata.uea.ac.uk/cru/data/parislondon/), cloud cover and direction as well as temperature from
Paris is available from Climate of the Past (Pliemon et al., 2022). Documentary data are available from
https://boris-portal.unibe.ch/handle/20.500.12422/207 (Burgdorf et al., 2022).
**Conflict of interest statement:** The author declares no conflict of interests.
**Author contributions:** SB digitized the diary together with a student, performed all analyses and wrote the
paper.

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
