# Peer review of "The weather diary of Georg Christoph Eimmart for Nuremberg, 1695-1704"

_Climate of the Past, 2022_

## Author Response (AR1)

**Reply to Reviewer 1**

The paper analyses the content of the weather diary of GG Eimmart for Nuremberg, 1695-1704. The content is both symbolic and literal, evolves through time and the final and more stable version uses a limited number of symbols. The value of weather diaries as a source of meteorological and climatological information has been previously demonstrated in the literature and this paper adds a new piece to the incomplete puzzle of the Late Maunder Minimum weather/climate.

The paper is well organised and easy to read. The methods to extract the information are well described and statistics of the results are provided. As is usually the case, the author provides some case studies to support the consistency of the data obtained from the diary with those obtained from other sources. Therefore, I think that the paper is suitable for publication in CP after some minor changes.

The discussion of the Winter 1697-8 would benefit if evidences of the circulation over the Atlantic contained in a recent paper by Mellado-Cano et al 2020.

Thanks, a brief analysis is added. The patterns in Fig .9 resemble those of the Easterly Index in Mellado-Cano et al. 2020, and in fact easterly wind components are found in the English Channel winds, in Paris cloud motion, and winds at Nuremberg.

In my opinion, the author uses the terms spatial (lines 364-365) and temporal information (lines 419-421) in a misleading way. The information from a single point can only be temporal and you can only obtain spatial information aggregating several points. I recommend rephrasing the sentences where those terms are associated.

Thanks. I know specify this better. Both instances refer to all weather data, not just the Eimmart data. I now refer more explicitly to "combination of all weather data" in the first and "information contained in all available data sources" in the second instance.

Minor:

Line 33 'based instrumental data' should be 'based on instrumental data' Thanks

Line 50 a reference to the use of new approaches could be appropriate here Thanks. There is currently not much out there. But I add a reference to Philip Brohan's text in this, although this is not a journal article.

Brohan, P. (2022) Machine Learning for Data Assimilation. http://brohan.org/Proxy_20CR/

Line 85 what do you mean by larger observatory? Should be clarified According to Gaab (2005) it was the most well-known. This is now added (with the reference)

Line 258 Kaliningrad is repeated Thanks

**Reply to Reviewer 2**

I agree with the comments made by Reviewer #1 that this an interesting paper that is entirely suitable for publication in this journal. The author elegantly combines information about the nature of the diary, and the challenging interpretation of the values, with the potential significance of these and similar data in future reconstruction efforts. I have only a few minor comments that need addressing.

Minor comments

- Lines 9, 30 and 33: Please rephrase these sentences to make clear that it is the data from these weather diaries that is used, e.g. for line 9: "The data extracted from weather diaries have long been used..." Thanks, all three sentences are reformulated to "data from these weather diaries".

- Line 41: I think it is worth adding here that the manual reconstructions are also subjective and this can lead to problems. I'm thinking, for example, about the "decline in the westerlies" in the Lamb Weather Types that was found to be an artefact of the map-drawing when objective reconstructions were made (Jones et al. (2014) and references therein) Thanks, a sentence (and the reference) was added.

- Line 22: "Correctness" to "reliability" Thanks

- Line 307: Specify here that the EKF400 data are monthly resolution Thanks

Reference

Jones, P.D., Osborn, T.J., Harpham, C. and Briffa, K.R. (2014), The development of Lamb weather types: from subjective analysis of weather charts to objective approaches using reanalyses. Weather, 69: 128-132. https://doi.org/10.1002/wea.2255